# Thrombocytopenia and Therapeutic Strategies after Allogeneic Hematopoietic Stem Cell Transplantation

**DOI:** 10.3390/jcm11051364

**Published:** 2022-03-02

**Authors:** Leyre Bento, Mariana Canaro, José María Bastida, Antonia Sampol

**Affiliations:** 1Hematology Department, Son Espases University Hospital, 07120 Palma, Spain; mariana.canaro@ssib.es (M.C.); antonia.sampolm@ssib.es (A.S.); 2Institut d’Investigació Sanitària Illes Balears (IdISBa), 07120 Palma, Spain; 3Hematology Department, Complejo Asistencial Hospital Universitario de Salamanca (CAUSA), Instituto de Investigación Biomédica de Salamanca (IBSAL), Universidad de Salamanca (USAL), 37007 Salamanca, Spain; chema@usal.es

**Keywords:** allogeneic stem cell transplantation, thrombocytopenia, morbidity, thrombopoietin receptor agonists

## Abstract

Thrombocytopenia following allogeneic hematopoietic stem cell transplantation is a usual complication and can lead to high morbidity and mortality. New strategies, such as the use of another graft versus host-disease prophylaxis, alternative donors, and management of infections, have improved the survival of these patients. The mechanisms are unknown; therefore, the identification of new strategies to manage this potentially serious problem is needed. Thrombopoietin receptor agonists are currently available to stimulate platelet production. Some small retrospective studies have reported their potential efficacy in an allogeneic stem cell transplant setting, confirming good tolerability. Recent studies with higher numbers of patients also support their safety and efficacy in this setting, hence establishing the use of these drugs as a promising strategy for this post-transplant complication. However, prospective trials are needed to confirm these results.

## 1. Introduction

Thrombocytopenia following allogeneic hematopoietic stem cell transplantation (allo-SCT) is a usual complication and could often increase morbidity and mortality [1]. Mechanisms are usually multifactorial and poorly known [2]. Prolonged isolated thrombocytopenia (PT) has been described in 5–20% of cases [3] and is defined as persistent thrombocytopenia (<20,000/mm^3^) associated with normal count in other blood cell lines or the requirement of transfusion less than 60 days after allo-SCT [4]. It may be caused by antibodies, splenomegaly, or delayed production attributable to decreased megakaryocytic differentiation. Meanwhile, the secondary failure of platelet recovery (SFPR) proposed by the Seattle group has been defined as platelet counts to <20,000/mm^3^ for 7 consecutive days or requirement of transfusion after reaching platelets ≥50,000/mm^3^ without transfusion for 7 days post-SCT. It has been estimated to occur in approximately 20% of allo-SCT recipients [5]. Risk factors include hematopoietic stem cell (HSC) dose, HLA disparity between donor and recipient, intensity of the conditioning regimen, infections, immunosuppression, drugs or graft versus host disease (GVHD). Finally, poor graft function (PGF) is defined as persistent cytopenia in the presence of complete donor chimerism and can affect one or more lineage [6]. The treatment of these causes of thrombocytopenia is mainly based on platelet transfusion and/or growth factor support, with other options not clearly defined.

## 2. Causes of Thrombocytopenia after Allo-SCT

### 2.1. Prolonged Isolated Thrombocytopenia (PT)

PT is defined as the recovery of all other cell lines, but consistently low platelet counts after allo-SCT. Incidence differs from 5 to 20% of SCT depending on the degree and duration of thrombocytopenia (60 days to 6 months). Previous reports indicated that thrombocytopenia on day 60 after allo-SCT is an independent factor for bad prognosis [7].

The mechanism involved in the evolution of PT remains confused. On the one hand, normal platelets may be generated by the bone marrow (BM) but destroyed too soon in the peripheral circulation due to splenomegaly or autoantibodies. Diagnosis of immune thrombocytopenia (ITP) mediated by platelet autoantibodies is challenging after allo-SCT because various complications, such as GVHD, disease relapse, viral infection, thrombotic microangiopathy and drug side effects, can also cause thrombocytopenia. An assessment of reticulated platelets and plasma thrombopoietin levels may be useful to distinguish between ITP and hypoplastic thrombocytopenia. ITP is generally characterized by an increased percentage of reticulated platelets and a normal or slightly increased plasma thrombopoietin level [8]. On the other hand, platelets may not be generated in adequate numbers due to decreased megakaryocyte differentiation. Bielski et al. confirmed that the careful evaluation of megakaryocytes in BM biopsies at the beginning of allo-SCT might predict increased risk of PT [3].

Alterations in the BM microenvironment involved in the pathogenesis of PT additionally remain unknown. In a recent prospective case-control study [9], the authors demonstrated that BM T-cell response was abnormal, with high proportions in the BM microenvironment of Th17, Th1 and Tc1 cells; and high levels of Th17 and Th1-associated cytokines (IL-6, IL-17, IL-21 and IFN-y,) in plasma. Zhang et al. [10] demonstrated an increase in immature megakaryocytes in allo-SCT patients with PT with low ploidy, which was related to the recruitment of CD8 T-cells, suggesting the role of the BM immune microenvironment.

### 2.2. Secondary Failure of Platelet Recovery (SFPR)

The incidence of SFPR in patients receiving allo-SCT not caused by relapse or graft rejection was found to be 20% and was described as an important risk factor for mortality [5]. The reasons are normally multifactorial; therefore, determining them to prevent and treat SFPR in clinical practice is complicated. Several risk factors have been found, including GVHD prophylaxis, unrelated donor, renal or liver alterations; HSC dose, infections and conditioning regimen with a combination of total-body irradiation, busulfan and cyclophosphamide. Other potential causes are immune-mediated thrombocytopenia, thrombotic microangiopathy, and drugs such as ganciclovir. 

New transplantation strategies, such as use of alternative donors, reduced-intensity conditioning regimens, GVHD prophylaxis, and the management of infections, have been improved. These advances and improvements might influence the mechanism related to thrombocytopenia and probably change the incidence and risk factors. Akahoshi et al. confirmed that the 3-year-cumulative incidence (CI) of SFPR was 12.2%, which is lower than previously reported [2]. One possible explanation could be the decrease in the incidence of cytomegalovirus (CMV) disease and advances in its treatment. The author also confirmed that acute GVHD and ATG or alemtuzumab use were associated with a higher risk of SFPR. Possible mechanisms leading to low platelet counts in patients with GVHD may include reduced platelet production and increased consumption [11]. 

A preemptive approach based on improved CMV techniques has reduced the incidence of CMV disease, which is an important potential cause of SFPR, and the cumulative doses of valganciclovir or ganciclovir, which could generate myelotoxicity. Meanwhile, new antivirals such as letermovir, an agent that inhibits CMV replication, could be a new option to reduce toxicity. Marty et al. confirmed that letermovir prevented clinically significant infection in CMV-seropositive recipients. Its use was highly effective, beginning at a median of 9 days after allo-SCT and administered at week 14 (approximately day 100 after allo-SCT), and was related to lower mortality at week 24 after transplantation [12]. The absence of myelotoxicity allowed for the beginning of letermovir prophylaxis before engraftment.

### 2.3. Poor Graft Function (PGF)

PGF after allo-SCT is another entity with no standard criteria for diagnosis. It usually refers to persistent cytopenia in more than one cell lineage, accompanied by a hypoplastic/aplastic bone marrow and complete donor chimerism in the absence of relapse [13]. There is usually a dependence on blood and/or platelet transfusions and/or growth factor support in the absence of other explanations such as disease relapse, drugs or infections [14]. 

The etiology of PGF after allo-SCT is understood. Various pre-transplant causes (unrelated donor, liver or renal dysfunction, infections), peri-transplant (conditioning regimens including total body irradiation, HSC dose, T-cell depletion, GVHD prophylaxis such as methotrexate) and post-transplant complications (GVHD, viral reactivations or drugs) have been associated not only with SFPR but also with PGF after allo-SCT [11,13]. It is thought that these causes trigger or contribute to an inflammatory and immune microenvironment, which represents the pathogenic mechanism of PGF. After corrections of all reversible causes, treatment options for these entities are limited. 

Poor survival has been reported for patients with PGF compared with those with good engraftment. In a case-control study of 830 patients reported by Zhao and colleagues [15], 24 patients (3%) developed PGF with very poor outcomes compared with patients with normal graft function (1-year overall survival, 25% versus 91%). Low graft cell dose (<5 × 10^6^ CD34 cell/kg), high ferritin (>2000 ng/mL) and splenomegaly were associated with a higher risk of primary PGF. However, no associations with donor HLA type or conditioning regimens were observed. 

## 3. Management of Post-Transplant Thrombocytopenia

Treatment options for thrombocytopenia after allo-SCT are unclear and are generally based on transfusion. Nevertheless, transfusion support is related to several adverse effects, such as platelet refractoriness, infusion reactions, acute lung and cardiac injury, with a final, heavy, financial burden. The trigger level for prophylactic platelet transfusion has been established as 10,000/mm^3^. A prospective randomized trial of a prophylactic platelet transfusion trigger of 10,000/mm^3^ versus 30,000/mm^3^ in allo-SCT recipients confirmed that a trigger level of less than 10,000/mm^3^ considerably reduced the number of platelet transfusions and the cost, without increasing the incidence or the severity of hemorrhagic events [16]. However, Diedrich and colleagues [16] concluded that additional, randomized, controlled clinical trials are needed to answer the open question of whether prophylactic platelet transfusions can be replaced by therapeutic ones in patients in a stable condition. 

CD34 selected stem cell boost (SCB) has also been used for patients with insufficient engraftment including severe thrombocytopenia. Shahzad et al. recently published a systematic review and meta-analysis including 209 patients who received CD34 selected SCB for PGF after allo-SCT. Primary graft sources included peripheral blood stem cells (72%) and bone marrow (28%). The median time from allo-SCT to SCB was 138 days (range 113–450) and the median SCB dose 3.45 × 10^6^ CD34 cell/kg (range 3.1–4.9). Complete response and overall response rates were 72% (95% CI, 63–79%) and 80% (95% CI, 74–85%), respectively [17]. Regarding CD3 T cells, a lower number (median 1.27 × 10^3^ cells/kg) was reported by Mohty et al. who did not find any cases of GVHD. This supports the idea that the CD3 T cells dose in CD34 selected SCB correlates with the incidence of GVHD [18]. Unfortunately, prospective studies are needed to confirm the optimal dose and manipulation of SCB.

The use of expanded mesenchymal stem cells (MSCs) as a potential alternative to SCB has been recently investigated. MSCs differ from SCB in the fact that they do not need to be collected from the primary allo-SCT donor but instead can be collected from a third person. Liu and colleagues [19] reported a series of 20 patients treated with MSCs: 17 experienced hematopoietic recovery but 13 developed infections, including 7 EBV, 3 of whom developed EBV-associated post-transplantation lymphoproliferative disorder. 

These procedures, as mentioned before, are not always exempt from potential risks, and can be inaccessible. Therefore, it is essential to recognize new tools to manage this complication. At present, thrombopoietin receptor agonists (TPO-RAs) that activate platelet production are available and could be promising options in this setting. To date, fewer than 30 retrospective studies have analyzed the probable benefits of TPO-RAs in post-SCT settings and only included a few patients. Further, there are already six ongoing clinical trials in this setting (www.clinicaltrials.gov, accessed on 17 January 2022). the results from these trials are not yet available, excepting a preliminary description from a phase II trial [20]. In this study, 60 (53 allogeneic and 7 autologous) transplanted patients with persistent thrombocytopenia or neutropenia treated with eltrombopag were enrolled. The response rate was 36%, but results were statistically inconclusive in comparison with the control arm in terms of superiority (28% responses).

### 3.1. Eltrombopag

Approved for the management of refractory ITP [21,22] and thrombocytopenia secondary to hepatitis C infection [23], this has also been associated with patients with refractory aplastic anemia [24] with multilineage responses, supporting the direct stimulation of the surviving HSC [25]. Experience with eltrombopag in an allo-SCT setting is summarized in Table 1.

Fu et al. recently reported one series of 38 patients after haploidentical SCT who received eltrombopag [33], and the CI of overall response (ORR) was 63.2%. Fifteen patients had SFPR, 15 poor graft function and 8 delayed engraftments. In addition, similar results with a lower number of patients (n = 13) were published by Yuan and colleagues (ORR of 62%) [32]. Not long ago, a Spanish group confirmed that PT required a longer response time in comparison with SFPR: 93 days (8–217) compared to 60 days (2–247), respectively [34]. Nonetheless, Fu and colleagues [33] confirmed a shorter response time, 17 days (2–89) compared to 66 (2–247), in the Spanish group experience [34]. During treatment, around 15% of patients developed liver function alterations (elevated transaminases >2.5 times or bilirubin twice normal levels), but no patient stopped the treatment because of adverse effects or intolerability [33,34].

Eltrombopag induces the differentiation of CD34+ cells into CD41+ megakaryocyte progenitor cells [35]. In addition, this drug stimulates the c-MPL receptor and can improve hematopoiesis cells (erythroid, platelet and neutrophil lines) [24,36,37]. In the recent retrospective experience presented by a Spanish group [34], 25% of the patients had <1000/µL neutrophils prior to TPO-RAs and, of these, 77% achieved ≥1000/µL after treatment. Regarding erythroid cells, 12% had hemoglobin <8 g/dL prior to TPO-RAs and half of them achieved ≥8 g/dL after eltrombopag. However, we could not rule out that the hematological response could be at least partially responsible for the resolution or correction of the possible causes of cytopenia, such as GVHD or infections. Deaths were lower in responder-patients to TPO-Ras: 15% versus 53% in non-responders (*p* < 0.001) [34]. This confirms that refractory thrombocytopenia is a poor prognostic parameter for survival in an allo-SCT setting and probably a surrogate factor of an altered immune system or defective hematopoiesis.

Regarding megakaryocyte numbers as a predictor of response, Bento et al. showed that 81% of patients had a decreased number prior to TPO-Ras, showing a slower response to treatment: median time to ≥20,000/mm^3^ platelets 43 versus 28 days (*p* = 0.019) [34]. Tanaka et al. also reported 12 patients treated with TPO-Ars, with a higher and faster platelet recovery in those with normal megakaryocytes before TPO-Ras than in those with decreased megakaryocytes. These results suggest that the presence of megakaryocytes in BM may better predict the response to these agents than the type of thrombocytopenia after transplant [28].

### 3.2. Romiplostim

Romiplostim is approved for refractory chronic ITP [38]. Both romiplostim and eltrombopag show promising effects in the management of myelodysplastic syndrome (MDS)-related thrombocytopenia [39,40]. Some studies with a small number of cases (<10 patients) have published the effects of romiplostim in allo-SCT recipients. Hartranft and colleagues published the largest study to date on the use of romiplostim in 13 patients with thrombocytopenia after allo-SCT [41]. In this series, 54% achieved the main endpoint of platelets ≥ 50,000/µL with a median of 35 (range 14–56 days) following the initiation of TPO-RAs. Experience with romiplostim after allo-SCT is summarized in Table 2. Data from clinical trials are not accessible, except a recent preliminary report from a phase I trial including 20 patients, which confirmed 10 mcg/kg/dose as the maximum tolerated dose of the drug in this setting [42].

Contrary to eltrombopag, whose experience has been described to induce a multi-lineage response in refractory aplastic anemia [24,43], there is a paucity of data about the effect of romiplostim in multi-lineage responses [44]. Few data are available to understand such differences, considering that the same molecular pathways are activated by both agonists.

## 4. Materials and Methods

This review includes 20 retrospective studies showing the experience of TPO-RAs in thrombocytopenia (PT, SFPR or PGF) after allo-SCT (Table 1 and Table 2), published from 2010 to date. Three independent authors reviewed the manuscript. The experience of TPO-RAs in thrombocytopenia after autologous SCT was excluded. Unfortunately, most results from clinical trials regarding the use of TPO-RAs in an allo-SCT setting have not yet been published.

## 5. Conclusions

Thrombocytopenia after allo-SCT is a usual complication and a challenge in our daily practice. Although it can lead to increased morbidity and mortality, new transplantation strategies, such as the use of GVHD prophylaxis, alternative donors and management of infections, have improved the survival of these patients. Recent studies showing the experience of TPO-RAs for thrombocytopenia after allo-SCT, support their safety and efficacy in this new setting, with a low number of side effects. However, further prospective trials are needed in order to identify the predictors of response and increase the level of evidence.

## Figures and Tables

**Table 1 jcm-11-01364-t001:** Experience with Eltrombopag for persistent thrombocytopenia in an allo-SCT setting.

Reference	Year	Thrombocytopenia Type	N	TPO-RAs	Transfusion Independence	Response Rate Platelets ≥ 50 × 10^9^/L
Reid et al., AJH [26]	2012	PT	1	Eltrombopag	Yes	NR
Fujimi et al., Int J Hematol [27]	2015	PT	1	Eltrombopag	Yes	1/1 (100%)
Tanaka et al., BBMT [28]	2016	PT, SFPR	12	Eltrombopag	Yes (n = 9)	9/12 (75%)
Bosch-Vilaseca et al., EJH [29]	2018	PT, SFPR	20	Eltrombopag Romiplostim	Yes (n = 12)	12/20 (60%)
Rivera et al., BMT [30]	2018	PT, SFPR	14	Eltrombopag	Yes	8/14 (57%)
Marotta et al., BMT [31]	2018	PT, SFPR	13	Eltrombopag	Yes	7/13 (54%)
Yuan et al., BBMT [32]	2019	PT, SFPR	13	Eltrombopag	Yes (n = 8)	8/13 (62%)
Fu et al., BMT [33]	2019	PT, SFPR	38	Eltrombopag	Yes (n = 24)	24/38 (63%)
Bento et al., BBMT [34]	2019	PT, SFPR	86	Eltrombopag Romiplostim	NR	62/86 (72%)
Nampoothiri et al., BMT [13]	2021	PT, SFPR, PGF	17	Eltrombopag	NR	10/17 (59%)
Giammarco et al., Int J Hematol [6]	2021	PGF	48	Eltrombopag	Yes (n = 36)	24/48 (50%)

TPO-RAs: Thrombopoietin receptor agonists; PT: Prolonged isolated thrombocytopenia; SFPR: Secondary failure of platelet recovery; PGF: Poor graft function; NR: Not reported.

**Table 2 jcm-11-01364-t002:** Experience with Romiplostim for persistent thrombocytopenia in an allo-SCT setting.

Reference	Year	Thrombocytopenia Type	N	TPO-RAs	Transfusion Independence	Response Rate Platelets ≥ 50 × 10^9^/L
Beck et al., Pediatr Blood Cancer [45]	2010	SFPR	1	Romiplostim	Yes	1/1 (100%)
Calmettes et al., BMT [46]	2011	SFPR	7	Romiplostim	Yes	7/7 (100%)
Bollag et al., EJH [47]	2012	SFPR	1	Romiplostim	Yes	1/1 (100%)
Poon et al., Am J Blood Res [48]	2013	PT, SFPR	3	Romiplostim	Yes	3/3 (100%)
DeRemer et al., J Oncol Pharm Practice [49]	2013	SFPR	1	Romiplostim	No	NR
Buchbinder et al., Pediatr Transplantation [50]	2015	SFPR	1	Romiplostim	Yes	1/1 (100%)
Maximova et al., Int J Hematol [51]	2015	SFPR	7	Romiplostim	Yes (n = 6)	6/7 (86%)
Battipaglia et al., BMT [52]	2015	SFPR	3	Romiplostim	Yes	3/3 (100%)
Hartranft et al., J Oncol Pharm Practice [41]	2015	PT, SFPR	13	Romiplostim	Yes (n = 7)	7/13 (54%)

TPO-RAs: Thrombopoietin receptor agonists; PT: Prolonged isolated thrombocytopenia; SFPR: Secondary failure of platelet recovery; NR: Not reported.

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
