# Peer review of "Thrombocytopenia and Therapeutic Strategies after Allogeneic Hematopoietic Stem Cell Transplantation"

_jcm, 2022, doi:10.3390/jcm11051364_

Round 1

Reviewer 1 Report

Bento L. and colleagues herein report a Review on Thrombocytopenia after Allogeneic Stem Cell Transplant. The purpose of the review might be of interest for the readers, however the manuscript is affected by major limitations:

  • "Thrombocytopenia after HSCT" is way too general issue, as many different causes might occurr which need to be addressed properly The Review is not extensive, extremely short, and many issues (i.e. Poor graft function which represents a main cause of Thrombocytopenia after HSCT) are not discussed
  • Style is poor

Author Response

REVIEWER 1:

Bento L. and colleagues herein report a Review on Thrombocytopenia after Allogeneic Stem Cell Transplant. The purpose of the review might be of interest for the readers, however the manuscript is affected by major limitations:

  • "Thrombocytopenia after HSCT" is way too general issue, as many different causes might occur which need to be addressed properly. The Review is not extensive, extremely short, and many issues (i.e. Poor graft function which represents a main cause of Thrombocytopenia after HSCT) are not discussed. Style is poor.

Answer:

Thank you very much for your comments that have helped to improve our manuscript. We have extended the review article including these important issues such as poor graft function or other treatment strategies for thrombocytopenia after allogeneic hematopoietic stem cell transplantation (CD34 stem cell boost or mesenchymal stem cells). All additional information in the manuscript have been marked in red colour.

Reviewer 2 Report

This is wellwritten and comprehensive review of specific therapies for trombocytopenia following HCT.
However,I would like to see this review expanded by also covering some paragraphs with randomized studies on trigger level for platlet transfusions in HCT patients,the role of the graft ie bone marrow vs PBSC vs cord blood grafts,the role of platlet antibodies,the role of GVHD on trombocytopenia,booster graft effects on trombocytopenia and treatment of platlet refractory patients with novel approaches such as mesenchymal stromal cells and more.

Author Response

REVIEWER 2:

This is wellwritten and comprehensive review of specific therapies for trombocytopenia following HCT. However, I would like to see this review expanded by also covering some paragraphs with:

  • Randomized studies on trigger level for platelet transfusions in HCT patients.

Answers:

Thank you very much for your comment; this information is now added to the paper (page 3, paragraph 5 and line 4).

  • The role of the graft ie bone marrow vs PBSC vs cord blood grafts.

Answers:

We strongly agree that this information would be very interesting but unfortunately the majority of the studies included patients who received PBSC graft. No differences have been confirmed in any of these studies.

  • The role of platelet antibodies, the role of GVHD on trombocytopenia, booster graft effects on trombocytopenia and treatment of platelet refractory patients with novel approaches such as mesenchymal stromal cells and more.

Answers:

Many thanks for your comment that has helped to improve our manuscript. We have included the role of platelet antibodies (page 2, paragraph 2 and line 3), the role of GVHD on thrombocytopenia (page 2, paragraph 5 and line 9) and other approaches such as CD34 stem cell boost and mesenchymal stem cells in the text as suggested (page 3, paragraph 6 and line 1).

Round 2

Reviewer 1 Report

Unfortunately the revision performed by the authors is not extensive as needed, the manuscript remains still too vague for readers in the field of internal medicine with a hematology strong background. Further, major typo errors are present, as row 30 "less than 60 days after allo-SCT" maybe the authors meant "beyond 60 days"

Reviewer 2 Report

This review is substantially improved.It is informative and easy to read.

This manuscript is a resubmission of an earlier submission. The following is a list of the peer review reports and author responses from that submission.